# Sex differences in physical performance by age, educational level, ethnic groups and birth cohort: The Longitudinal Aging Study Amsterdam

**Lena D. Sialino**[1]*, **Laura A. Schaap**[1], **Sandra H. van Oostrom**[2], **Astrid C. J. Nooyens**[2], **Hendrika S. J. Picavet**[2], **Johannes W. R. Twisk**[3], **W. M. Monique Verschuren**[2,4], **Marjolein Visser**[1], **Hanneke A. H. Wijnhoven**[1]

1 Department of Health Sciences, Faculty of Science, Amsterdam Public Health research institute, VU University Amsterdam, Amsterdam, the Netherlands, 2 Centre for Nutrition, Prevention and Health Services, National Institute for Public Health and the Environment, Bilthoven, the Netherlands, 3 Department of Clinical Epidemiology and Biostatistics, VU University Amsterdam, Amsterdam, the Netherlands, 4 Julius Centre for Health Sciences and Primary Care, University Medical Centre, Utrecht, The Netherlands

* l.d.sialino@vu.nl

**Data Availability Statement:** Data cannot be shared publicly because of confidentiality. Data are available from the LASA Institutional Data Access /

## Abstract

### Background

Older women perform consistently poorer on physical performance tests compared to men. Risk groups for this "female disadvantage" in physical performance and it's development over successive birth cohorts are unknown. This is important information for preventive strategies aimed to enhance healthy aging in all older women. This study aims to longitudinal investigate whether there are risk groups for a more apparent female disadvantage and study its trend over successive birth cohorts.

### Methods

Data of the Longitudinal Aging Study Amsterdam (LASA) were used. All participants were aged 55–65 years at baseline. Longitudinal data of two birth cohorts with baseline measurements in 1992/1993 (n = 966, 24 year follow-up) and 2002/2003 (n = 1002, 12 year follow-up) were included. Follow-up measurements were repeated every three/four years. Cross-sectional data of two additional cohorts were included to compare ethnic groups: a Dutch cohort (2012/2013, n = 1023) and a Migration cohort (2013/2014, n = 478) consisting of migrants with a Turkish/Moroccan ethnicity.

### Results

Mixed model analysis showed that women aged 55 years and older had a lower age- and height-adjusted gait speed (-0.03 m/s; -0.063–0.001), chair stand speed (-0.05 stand/s; -0.071–-0.033), handgrip strength (-14,8 kg; -15.69–-13.84) and balance (OR = 0.71; 0.547–0.916) compared to men. The sex difference in handgrip strength diminished with increasing age, but remained stable for gait speed, chair stand speed and balance. In

Ethics Committee (contact via https://www.lasa-vu. nl/index.htm) for researchers who meet the criteria for access to confidential data. The data underlying the results presented in the study are available from the Longitudinal Aging Study Amsterdam (https://www.lasa-vu.nl/index.htm). In the analysis proposal the following variables with file codes should be included: Gait speed, chair stand and tandem balance test in file LASA034, handgrip strength and height in file LASA161, education and sex in file Z004 and age in file Z008. For each variable all the waves should be included, from all baseline waves (B) to the wave (I). The LASA Steering Group will review all requests for data to ensure that proposals for the use of LASA data do not violate privacy regulations and are in keeping with informed consent that is provided by all LASA participants. The authors of this study do not have any special access privileges to the data underlying this study that other researchers would not have.

**Funding:** This work was supported by the Netherlands Organization for Health Research and Development (ZonMw) [project number 849200005] (https://www.zonmw.nl/nl/). The Longitudinal Aging Study Amsterdam is supported by a grant from the Netherlands Ministry of Health Welfare and Sports, Directorate of Long-Term Care (https://www.government.nl/ministries/ministry-of-health-welfare-and-sport). The data collection in 2012-2013 and 2013-2014 was financially supported by the Netherlands Organization for Scientific Research (NWO) in the framework of the project "New Cohorts of young old in the 21st century" [file number 480-10-014] (https://www. nwo.nl/). The authors are grateful to all LASA participants for their valued contributions and have no conflict of interest to declare. The funders had no role in study design, data collection and analysis, decision to publish, or preparation of the manuscript.

**Competing interests:** The authors have declared that no competing interests exist.

general, results were consistent across different, educational levels and Turkish/Moroccan ethnic groups and birth cohorts.

## Conclusions

There is a consistent "female disadvantage" in physical performance among older adults, which remains stable with increasing age (except for handgrip strength) and is consistent across different educational levels, ethnic groups and successive birth cohorts. So, no specific risk groups for the female disadvantage in physical performance were identified. Preventive strategies aimed to enhance healthy aging in older women are needed and should target all older women.

## Introduction

Physical performance is an important indicator of healthy aging, since it is associated with a wide range of adverse health outcomes among older adults [1–4]. Women have a higher life expectancy but, paradoxically, older women perform poorer on physical performance tests than men [5–7]. This "female disadvantage" is well established, but its longitudinal course by age remains unclear. Furthermore, risk groups with regard to educational level and ethnic groups and trends over birth cohorts are unknown.

Most studies investigating the sex difference in physical performance by age are limited by a cross-sectional design or a short-term longitudinal follow-up period (five years) [5,8,9]. In exception, the well-studied sex difference in handgrip strength diminishes by age due to a more rapid decline in men compared with women [7,10,11]. There is some evidence from longitudinal data (ten years follow-up time) that the sex difference in gait speed remains stable during aging [12]. However, longitudinal research with a longer follow-up time is needed to confirm this finding, which will determine whether there is a specific age group where the female disadvantage is most apparent.

The influence of other sociodemographic factors on the sex difference in physical performance have not been studied so far. Low educated subgroups have a poorer physical performance [13], which is mainly explained by a higher prevalence of common chronic disease, obesity, smoking and though workload [14]. Since men and women differ in their lifestyle and incidence of chronic diseases [15], the influence of education on physical performance may differ between men and women. Indeed, more years of education seems to be a stronger and longer protective factor at higher ages, for a decline in gait speed for men compared with women [12]. More longitudinal research and other physical performance indications are needed to confirm this finding.

Ethnicity also influences physical performance, where persons with a European-American ethnicity have a higher gait and chair stand speed and better balance compared with persons with an African-American ethnicity [16,17]. This was largely explained by differences in education, obesity and diabetes [16]. Since men and women also differ in these factors [18], sex differences may differ per ethnic group. So far, this has not been investigated, but it was demonstrated that the sex difference in a physical performance summary score differs between some ethnic groups [16]. However, this research was limited to Hispanic- and Mexican-Americans ethnic groups only. The three largest ethnic groups in the Netherlands; Turkish, Moroccan and Dutch have not been studied so far. Since we know Turkish en Moroccan ethnic

groups in the Netherlands have a lower education, this might result in a larger sex difference compared to a Dutch ethnic group [19].

Since a female disadvantage in physical performance in older adults has been reported by different studies over the past three decades, it may be a consistent phenomenon throughout time. However, factors affecting healthy aging change over time, were recent cohorts of older adults have a healthier lifestyle, higher socio-economic status, self-reported health and life expectancy but more chronic diseases [20–22]. These birth cohort effects may differ between men and women, since older women of a recent cohort have a better health profile related to cardiovascular diseases and diabetes mellitus and their average educational level becomes higher compared to men [23]. Since these are known protective factors for a decline in physical performance, we expect a decrease in the female disadvantage over successive birth cohorts. However, longitudinal research is needed to confirm this hypothesis.

A more detailed understanding of the sex difference in physical performance with regard to the longitudinal course by age, educational level and ethnic groups may identify potential risk groups for a more pronounced female disadvantage. This will help determine target groups for preventive strategies to enhance healthy aging in all older women. Also, knowledge with regard to the course of the sex difference over successive birth cohorts will identify the current trend over time, informing us whether this is an increasing or decreasing phenomenon. This study aimed to longitudinally investigate risk groups for a more apparent female disadvantage and study its trend over successive birth cohorts, which was achieved.

## Methods

### Study population

Data from the prospective Longitudinal Aging Study Amsterdam (LASA) were used. This study was initiated to determine predictors and consequences of aging and contains a nationally representative sample from three culturally distinct regions in the Netherlands (Amsterdam, Zwolle and Oss). Measurements and interviews are performed by trained interviewers, who visit respondents at home [24]. For a more detailed description of LASA and the used questionnaires, see Huisman *et al.* (2011). Ethical approval for the LASA study was given by the Medical Ethics Committee of the VU University Medical Centre Amsterdam, and all participants provided written informed consent.

### (Birth) cohort populations

Longitudinal data of the birth cohort 1927–1937 with baseline measurements in 1992/1993 (n = 966, 25 year follow-up) and the birth cohort 1937–1947 with baseline measurements in 2002/2003 (n = 1002, 12 year follow up) were used. Follow-up measurements were performed every three to four years. Furthermore, cross-sectional data of an additional birth cohort (n = 1023) and a migration cohort including Turkish and Moroccan migrants (n = 478) both measured in 2012–2014 were used. All participants were aged 55–65 years at baseline.

### Physical performance

Four measures of physical performance were included in this study: gait speed, chair stand speed, handgrip strength and balance. Gait speed was measured by asking participants to walk 3 meters, turn around and walk 3 meters back as quickly as possible, recorded by trained staff using a stopwatch. Gait speed was expressed in meters per second. Chair stand speed was measured by asking participants to fold their arms across their chest and to stand up and sit down from a sitting position five times at usual pace. Chair stand speed was defined as the number

of chair rises per second. Handgrip strength was measured by asking participants to perform two maximum strength measurements for both hands in standing position with their arms along their body, recorded in kg by a Takei TKK 5001 dynamometer. Handgrip strength was defined as the average of the maximum measurement of each hand. In the migration cohort only the right hand was measured twice, where handgrip strength was defined as the maximum measurement. Balance was measured by asking participants to maintain their feet in the tandem position (heel of one foot directly in front of and touching the toes of the other foot) for ten seconds. Balance was defined as able if the test was performed correctly for at least 10 seconds, or unable if the test could not be performed due to physical inability or if balance was kept for less than 10 seconds.

## Birth cohort, educational level and ethnic groups

Two birth cohorts were included in the longitudinal analysis: birth cohort 1927–1937 and birth cohort 1937–1947. Educational level was categorized into: low (elementary education or less), middle (lower vocational education and general intermediate education) and high (intermediate vocational education, general secondary education, higher vocational education, college education and university). Ethnic groups were categorized as Dutch or Turkish/Moroccan, using data of birth cohort 1947–1957 and the migration cohort. These two cohorts are comparable with regard to age (mean, SD), year of measurement and design (cross-sectional).

## Statistical analyses

To examine the longitudinal data of birth cohorts 1927–1937 and 1937–1947, mixed model analysis with a random intercept for the individual was used for the continuous outcomes: gait speed, chair stand speed and handgrip strength. Generalized estimating equations (GEE) with an exchangeable correlation structure and robust variance estimator were used for the dichotomous outcome: ability to perform the tandem balance test. Here, GEE was used because the regression coefficients are calculated as "population averaged", which fits best for our population study [25]. To examine the cross-sectional data of birth cohort 1947–1957 and the migration cohort, linear and logistic regression analyses were used.

The percentage missing values is similar between men and women and is relatively stable over different measurements and outcome measures (approximately 15% on average). S1 and S2 Tables show the number of participants included in each analysis. Mixed model analysis and GEE include all longitudinal data of the outcome, thereby preventing the loss of data and power [25]. Taken this into account and the fact that there are no further missing values in sex, age and baseline height, we assume no implications due to missing values in our analysis and a robust regression coefficient estimate.

Several analysis steps were taken: (1) The overall age- and height-adjusted sex difference in physical performance was investigated for all longitudinal and cross-sectional cohort populations separately ("sex" in a model with "age"). (2) The longitudinal course of the sex difference by age was investigated using the longitudinal data of the birth cohorts 1927–1937 and 1937–1947. Therefore, the interaction term age and sex: "age*sex" was used. If "age*age" or "age*age*sex" was significant, the quadratic regression coefficient in addition to the linear regression coefficient was included in the models throughout the rest of the analysis steps. This allowed a better prediction of the decline in physical performance measures during aging. (3) The modification of the age- and height-adjusted sex difference by educational level, ethnic groups or birth cohort was investigated for all longitudinal and cross-sectional cohort populations separately (interaction term with sex, in a model with "age" and "sex": "birth cohort*sex",

"education*sex" or "ethnicity*sex"). (4) The modification of the course of the sex difference by age, by educational level or birth cohort was investigated using the longitudinal data of birth cohorts 1927–1937 and 1937–1947 (interaction term with "sex*age": "birth cohort*sex*age" and "education*sex*age"). In other words, whether the course of the sex difference by age was different for different birth cohorts or educational levels. All analyses were adjusted for baseline height to estimate the true unadjusted sex difference in physical performance, since it was demonstrated that height partly explains the sex difference in physical performance measures, but is not part of the proposed causal pathway between physical performance and various health outcomes [26]. Body height was measured to the nearest 0.001 m using a stadiometer by a trained interviewer.

To visually depict the longitudinal course of the sex difference by age, a trend line for gait speed, chair rise speed and handgrip strength was plotted by age and sex, based on the regression coefficients of the full model. To depict balance, the probability to be able to perform the tandem balance test was predicted by age and sex. All analyses were performed using SPSS (version 26.0, SPSS inc.).

## Results

### (Birth) cohort populations

All (birth) cohort populations had an equal amount of men and women, with a similar mean age at baseline (Table 1). Overall, women were lower educated, had a poorer physical performance and were more often unable to perform the physical performance tests due to physical problems compared to men.

### Age- and height-adjusted sex difference in physical performance

Adjusted for age and height, women performed in general poorer than men in gait speed, chair stand speed, handgrip strength and balance in both longitudinal cohorts, except for gait speed in birth cohort 1927–1937 where no significant sex difference was observed (Table 2). To illustrate, mean gait speed of women in birth cohort 1937–1947 was 0.05 m/s (95% confidence interval:-0.069–-0.021) lower than men. Also, women had about half the odds on the ability to successfully complete the tandem balance test for 10 seconds compared to men in birth cohort 1937–1947 (OR = 0.59, 95% confidence interval: 0.416–0.844). For birth cohort 1947–1957, women performed only significantly poorer in handgrip strength compared to men and for the migration cohort women performed significantly poorer in gait speed, chair stand and handgrip strength compared to men (Table 2).

### Longitudinal course of sex differences in physical performance by age

The sex difference in gait speed, chair stand speed and balance remained stable during aging for both birth cohort 1927–1937 and 1937–1947 (Fig 1) (Table 3). The first exception was handgrip strength, where the decline by age was larger in men compared to women for both birth cohorts (age*sex; p<0.001), resulting in a decrease of the sex difference with increasing age (Fig 1D) (Table 3). In addition, the sex difference in balance for birth cohort 1937–1947 slightly decreased by age in the exponential part of the decline in balance (sex*age*age; p = 0.001) (Table 3).

### Modification by educational level

For each educational level, women had a poorer physical performance compared with men (data not shown). In general, there was no significant difference between the educational levels

**Table 1. Baseline characteristics for men and women of the different (birth) cohort populations.**

| | Birth cohort 1927–37 | | Birth cohort 1937–47 | | Birth cohort 1947–57 | | Migration cohort 1948–58 | |
| --- | --- | --- | --- | --- | --- | --- | --- | --- |
| Characteristic variables | Men | Women | Men | Women | Men | Women | Men | Women |
| *Sociodemographic* | | | | | | | | |
| Study population | 467 [48] | 499 [52] | 475 [47] | 527 [53] | 496 [48] | 527 [52] | 275 [58] | 203 [42] |
| Age in years | 60 (2.8) | 60 (2.8) | 60 (2.9) | 60 (3.0) | 60 (2.9) | 60 (3.0) | 61 (3.1) | 61 (2.9) |
| Education | | | | | | | | |
| - Low education | 104 [22] | 202 [41] | 89 [19] | 124 [24] | 50 [9.3] | 55 [10] | 179 [65] | 164 [81] |
| - Middle education | 148 [32] | 191 [38] | 239 [50] | 334 [63] | 266 [54] | 328 [62] | 64 [23] | 26 [13] |
| - High education | 215 [46] | 106 [21] | 147 [31] | 69 [13] | 180 [36] | 144 [27] | 31 [11] | 8 [3] |
| *Physical performance* | | | | | | | | |
| Gait speed (m/s) Physically unable[a] | 0.96 (0.28) 5 [1.0] | 0.93 (0.28) 12 [2.4] | 1.0 (0.28) 6 [1.3] | 0.91 (0.26) 10 [1.9] | 1.1 (2.40) 2 [0.4] | 1.0 (0.24) 4 [0.8] | 0.76 (0.27) 3 [1.1] | 0.66 (0.27) 8 [3.9] |
| Chair stand speed (rise/s) Physically unable[a] | 0.45 (0.14) 19 [4.1] | 0.47 (0.45) 22 [4.4] | 0.49 (0.14) 11 [2.3] | 0.46 (0.13) 29 [5.5] | 0.43 (0.11) 8 [1.6] | 0.44 (0.11) 15 [2.8] | 0.39 (0.16) 20 [7.1] | 0.32 (0.12) 19 [9.4] |
| Handgrip strength (kg) Physically unable[a] | 40 (6.9) 4 [0.9] | 24 (4.9) 5 [1.0] | 45 (7.9) 13 [2.7] | 27 (5.8) 14 [2.7] | 44 (8.5) 1 [0.2] | 24 (6.2) 4 [0.8] | 36 (12.3) 3 [1.1] | 20 (8.2) 2 [1.0] |
| Balance[b] Physically unable[a] | 339 [73] 58 [12.4] | 350 [70] 100 [20.0] | 366 [77] 48 [10.1] | 376 [71] 80 [15.2] | 469 [95] 24 [4.8] | 472 [90] 51 [9.5] | 232 [84] 38 [13.8] | 136 [67] 52 [25.6] |

Birth cohort 1927–37 and 1937–47 contain longitudinal data, birth cohort 1947–57 and migration cohort contain cross-sectional data. Explanation of data: mean (SD), n [%].

[a]Participants who were unable to perform the test correctly/completely or refused due to being physical unable (for example; participants in a wheelchair).

[b]Number of participants able to successfully complete the tandem balance test (percentage of total).

in age- and height-adjusted sex difference or its course by age (S3 Table). In exception, for birth cohort 1937–1947 the sex difference in gait speed was lower in high educated participants compared to middle education persons (p = 0.017) but chair stand speed was higher in high and middle educated participants compared to low educated persons (respectively; p = 0.046 and p = 0.001).

## Modification by ethnic groups

Among older adults with a Turkish/Moroccan or Dutch ethnicity, women had a poorer physical performance compared to men (Table 2). The difference in age- and height-adjusted sex difference between the two ethnic groups was not consistent for different physical performance measures; the sex difference in chair stand speed was larger but in handgrip strength was smaller and for gait speed or balance similar in persons with a Turkish/Moroccan ethnicity compared to persons with a Dutch ethnicity. Noteworthy, the physical performance of both

**Table 2. Age- and height-adjusted mean difference in physical performance between women and men.**

| | Longitudinal data | | Cross-sectional data | |
| --- | --- | --- | --- | --- |
| *Physical performance* | Birth cohort 1927–1937 | Birth cohort 1937–1947 | Birth cohort 1947–1957 | Migration cohort 1948–1958 |
| Gait speed (m/s) B | -0.03 (-0.063–0.001) | **-0.05** (-0.069–-0.021) | -0.01 (-0.054–0.036) | **-0.08** (-0.140–-0.014) |
| Chair stand speed (rise/s) B | **-0.05** (-0.071–-0.033) | **-0.04** (-0.057–-0.028) | -0.02 (-0.039–0.001) | **-0.06** (-0.100–-0.027) |
| Handgrip strength (kg) B | **-14.8** (-15.69–-13.84) | **-16.3** (-17.01–-15.49) | **-14.8** (-16.10–-13.49) | **-15.5** (-18.01–-13.08) |
| Balance (ability) OR | **0.71** (0.547–0.916) | **0.59** (0.416–0.844) | 0.97 (0.929–1.022) | 0.96 (0.872–1.058) |

Abbreviations: B = regression coefficient, OR = odds ratio, both with 95% confidence interval. For example: the value of -0.05 means that the mean gait speed of women in birth cohort 1937–1947 is 0.05 m/s lower than men.

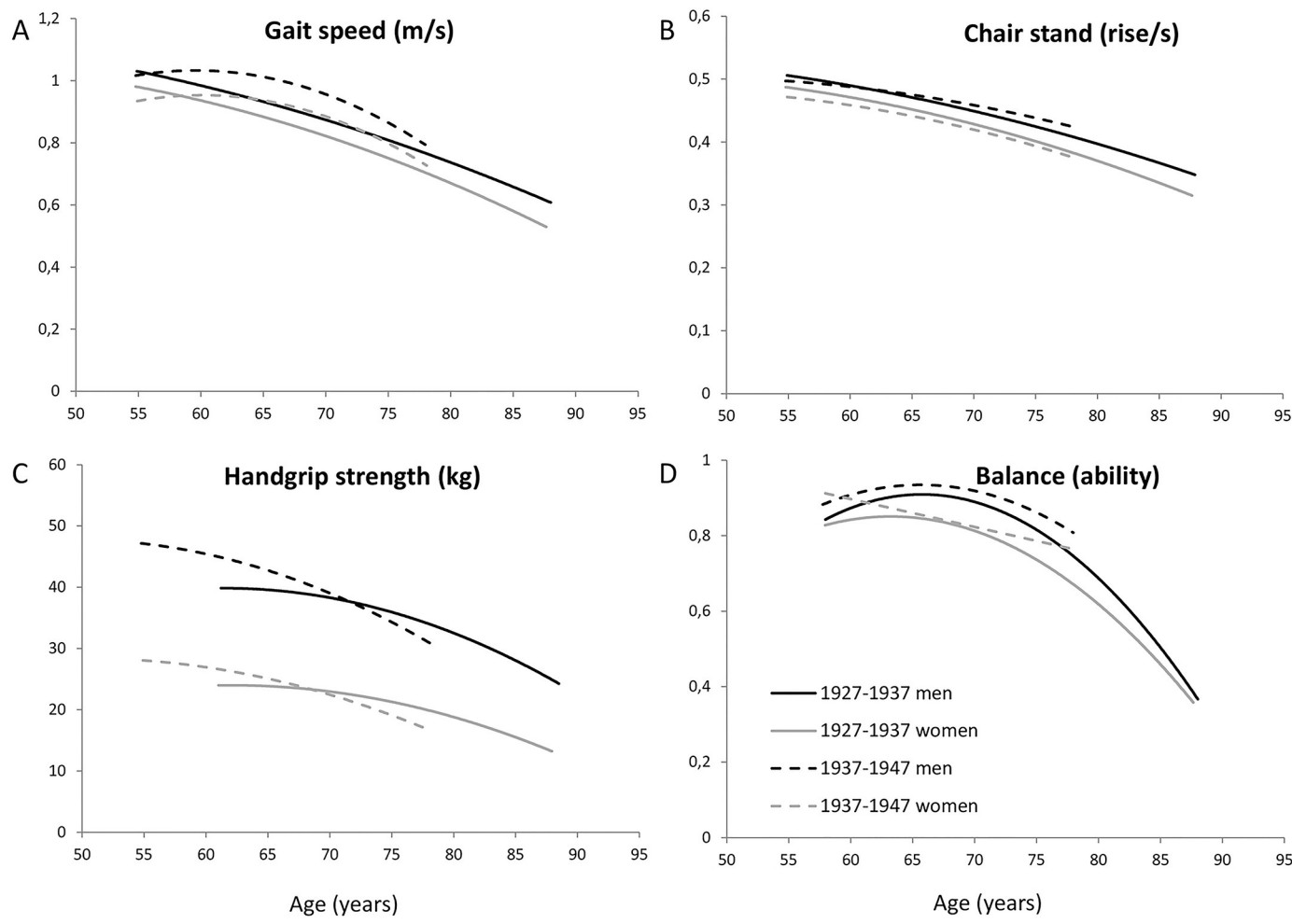

**Fig 1. Four physical performance measures by age and sex for two longitudinal birth cohorts 1927–1937 and 1937–1947.** Gait speed (A), chair rise speed (B), handgrip strength (C) and ability to perform the tandem balance test (D) by age for men and women. Handgrip strength in the 1927–1937 birth cohort was measured only from age 60 years and older. Balance is shown by the chance to be able to perform the tandem balance test, divided by one minus the chance to be able to perform the tandem balance test. Longitudinal data of birth cohort 1927–1937 (solid line) and 1937–1947 (dashed line).

men and women is poorer in Turkish/Moroccan ethnicities compared to persons with a Dutch ethnicity.

## Modification by birth cohort

There was no significant difference between the birth cohorts 1927–1937 and 1937–1947 in age- and height-adjusted sex difference in physical performance or its course by age (Fig 1) (S3 Table). The only exception was handgrip strength, where the age- and height-adjusted sex difference was slightly smaller for birth cohort 1927–1937 (-1.36 95% confidence interval; -0.166–-0.003 -, p = 0.026) and the sex difference decreased more rapidly by age compared with birth cohort 1937–1947 (p = 0.037) (Fig 1C) (S3 Table).

## Discussion

The present study confirmed a consistent age- and height- adjusted sex difference in physical performance in persons aged 55 years and older, were women perform worse compared to

**Table 3. Multivariate model of the longitudinal course of the height-adjusted sex difference in phsyical performance.**

| | Cohort | Gait speed | Chair stand | Handgrip strength | Balance[a] |
|---|---|---|---|---|---|
| Age (y) | 1927–37 | **-0.013 [-0.0139--0.1203], p<0.001** | **-0.005 [-0.0052--0.0045], p<0.001** | **-0.466 [-0.4971--0.4357], p<0.001** | **0.914 [0.902–0.926], p<0.001** |
| | 1937–47 | **0.009 [-0.0102--0.0073], p<0.001** | **-0.004 [-0.0043--0.0030], p<0.001** | **-0.581 [-0.6136--0.5548], p<0.001** | **0.953 [0.930–0.977], p<0.001** |
| + Age * Age (y²) | 1927–37 | **-0.001 [-0.0003--0.0001], p = 0.004** | **-0.001 [-0.0001--0.0001], p = 0.003** | **-0.018 [-0.0220--0.0131], p<0.001** | **0.967 [0.995–0.998], p<0.001** |
| | 1937–47 | **-0.001 [-0.0009--0.0005], p<0.001** | **-0.001 [-0.0002--0.0001], p = 0.037** | **-0.017 [-0.0210--0.0128], p<0.001** | 1.002 [0.994–1.002], p = 0.245 |
| + Sex (female) | 1927–37 | -0.028 [-0.0629–0.0007], p = 0.116 | **-0.052 [-0.0711--0.0330], p<0.001** | **-14.77 [-15.689--13.841], p<0.001** | **0.708 [0.547–0.916], p<0.001** |
| | 1937–47 | **-0.050 [-0.0781--0.0212], p = 0.001** | **-0.042 [-0.0569--0.0271], p<0.001** | **-16.29 [-17.079--15.492], p<0.001** | **0.592 [0.416–0.844], p = 0.004** |
| + Sex (female) * Age (y) | 1927–37 | -0.001 [-0.0034--0.0011], p = 0.505 | -0.001 [-0.0010–0.0005], p = 0.573 | **0.156 [0.0959–0.2160], p<0.001** | 1.011 [0.986–1.037], p = 0.951 |
| | 1937–47 | 0.001 [-0.0021–0.0036], p = 0.616 | -0.001 [-0.0023–0.0003], p = 0.115 | **0.201 [0.1443–0.2571], p<0.001** | 1.018 [0.931–1.036], p = 0.516 |
| + Sex (female) * Age * Age (y²) | 1927–37 | -0.001 [-0.0003--0.0001], p = 0.660 | -0.001 [-0.0035–0.0001], p = 0.476 | 0.006 [-0.0026–0.0151], p = 0.166 | 1.002 [0.999–1.006], p = 0.095 |
| | 1937–47 | 0.001 [-0.0003–0.0005], p = 0.555 | 0.001 [-0.00017–0.0002], p = 0.870 | 0.005 [-0.0035–0.0127], p = 0.263 | **1.010 [1.002–1.018], p = 0.011** |

Longitudinal data of LASA birth cohort 1927–1937 (light grey) and LASA birth cohort 1937–1947. Beta [95% confidence interval], p-value.

[a]OR [95% confidence interval], p-value.

men [1,7,27]. The sex differences for gait speed and handgrip strength are clinically relevant, because it reaches the minimally clinical significant individual difference estimates for gait speed (0.03–0.05 m/s) [28] and handgrip strength (5.0–6.5 kg) [29].

The female disadvantage in gait speed, chair stand speed and balance remained stable during aging, except for handgrip strength where it reduced by increasing age, confirming previous longitudinal findings for gait speed and handgrip strength [12,30]. In contrast, previous cross-sectional studies suggested an increase in the female disadvantage in gait speed and balance with age [7,8], demonstrating the importance of longitudinal studies.

The present study observed no consistent risk groups where the female disadvantage is most apparent, with regard to educational level and ethnic groups. The results of this study showed a better physical performance by participants with a Dutch ethnicity or higher education compared with other ethnicities (Moroccan and Turkish) or lower education, which is in line with previous research [13,16,17]. The current study demonstrated that these differences are similar for men and women.

This study is the first to show that the sex difference in physical performance and its longitudinal course by age does not significantly differ between the two investigated birth cohorts. Although the first birth cohort 1927–1937 was born before and birth cohort 1937–1947 born during world war two, this did not influence the sex difference in physical performance. Other factors that changed between the two birth cohorts, also did not influence the sex difference. This suggests that the female disadvantage is a robust phenomenon.

The observed findings raise the question whether the female disadvantage in physical performance during aging is due to a sex difference in physiology or in unhealthy aging. When the female disadvantage is present throughout the adult life course, it may suggest physiological differences between men and women, which may provide all women with a disadvantage

in physical performance. On a different note, men and women might perform at different physical performance levels, although they have the same health status. This would suggest the use of sex-specific cut-off points and remains to be investigated. Next to a physiological difference, the sex difference in physical performance may also arise due to differences in (un)healthy aging such as lifestyle and chronic diseases. The present study did not identify at which age the female disadvantage arises since it was already present at age 55, suggesting it could be a physiological sex difference. However, there was no sex difference demonstrated in similar physical performance tests among healthy adults aged 20 to 39 years [5,31,32], except for handgrip strength [33]. This may point towards sex differences in (un)healthy aging or to a ceiling effect of these tests. Previous research supports both theories or the combination [7]. For example, the higher body fat percentage in women has been suggested to put them at a significant biomechanical disadvantage for greater disability at older age [34], suggesting a sex difference in (un)healthy aging. In addition, sex differences in exposure to lifestyle factors and chronic health conditions in older adults were shown to be explanatory factors [7]. For example, women have a higher prevalence and perceived disease burden of osteoarthritis, which may explain the observed sex difference in physical performance [35,36]. On the other hand, sex differences in muscle strength and exposure to sex hormones were also identified as possible explanatory factors [7,34], suggesting a sex difference in physiology. Future longitudinal research across the adult life span investigating physiological and (un)healthy aging explanations is recommended.

The present study has several strengths. First, data from a large prospective longitudinal cohort study were used, which represents the older adult life course of the Dutch population [37]. To note, response rate was high and drop-out was low; approximately 3% per follow-up. Secondly, the longitudinal design made it possible to study the longitudinal course by age. Thirdly, the use of four different objectively physical performance measures provided a broad image but allowed the study of various aspects of physical performance separately, in contrast to a combined physical performance score [38]. Furthermore, all measures are strongly associated with negative health outcomes [1–4] and demonstrated to be of high quality [39]. At last, this is the first time that the influence of various sociodemographic factors are investigated in one large longitudinal study. A possible limitation is the loss to follow-up due to mortality, since healthy people live longer and women have a higher life expectancy then men (S2 Table) [11]. This could have caused an overestimation of the mean age- and height-adjusted sex difference, since more unhealthy men compared with unhealthy women are lost. In contrast, more women than men were physically unable to perform the physical performance tests, which may have caused an underestimation. Secondly, the influence of ethnic groups could only be tested in a cross-sectional setting.

In conclusion, this large prospective longitudinal cohort study showed that in persons aged 55 years and older there is a consistent and stable female disadvantage in physical performance which pertains by increasing age. There are no indications for specific risk groups, with regard to educational level and ethnic groups or for a change of this female disadvantage in the coming decades. Therefore, future research on underlying mechanisms, explanations and preventive healthy aging strategies aimed to reduce the female disadvantage, should target all older adults. This novel information forms a solid basis for future research and strategies regarding the female disadvantage in physical performance.

## Supporting information

**S1 Table. Number participants per follow-up (FU) measurement for longitudinal birth cohorts 1927–1937 and 1937–1947.** Percentage and number of participants for each physical

performance measurement (percentage of total at same follow-up measurement).
(DOCX)

**S2 Table. Number participants for each physical performance measurement (percentage of total) for cross-sectional birth cohorts 1947–1957 and migration cohort.**
(DOCX)

**S3 Table. Effect modification of sex differences in physical performance by education, ethnic groups and birth cohort.**
(DOCX)

## Acknowledgments

The authors are grateful to all LASA participants for their valued contributions.

## Author Contributions

**Conceptualization:** Astrid C. J. Nooyens, Hendrika S. J. Picavet, W. M. Monique Verschuren, Marjolein Visser.

**Formal analysis:** Lena D. Sialino.

**Investigation:** Lena D. Sialino.

**Methodology:** Lena D. Sialino, Johannes W. R. Twisk.

**Supervision:** Laura A. Schaap, Sandra H. van Oostrom, Hanneke A. H. Wijnhoven.

**Writing – original draft:** Lena D. Sialino.

**Writing – review & editing:** Lena D. Sialino, Laura A. Schaap, Sandra H. van Oostrom, Hanneke A. H. Wijnhoven.

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
