## [Decision Letter · Decision Letter 0]

20 Aug 2019

PONE-D-19-18300

Sex differences in physical performance by age, birth cohort, educational level and ethnic groups; The Longitudinal Aging Study Amsterdam

PLOS ONE

Dear Dr. Sialino,

Thank you for submitting your manuscript to PLOS ONE. After careful consideration by 2 Reviewers and an Academic Editor, all of the critiques of both Reviewers must be addressed in detail in a revision to determine publication status. If you are prepared to undertake the work required, I would be pleased to reconsider my decision, but revision of the original submission without directly addressing the critiques of the two Reviewers does not guarantee acceptance for publication in PLOS ONE. If the authors do not feel that the queries can be addressed, please consider submitting to another publication medium. A revised submission will be sent out for re-review. The authors are urged to have the manuscript given a hard copyedit for syntax and grammar.

**Comments to the Author**

1. Is the manuscript technically sound, and do the data support the conclusions?

Reviewer #1: Partly

Reviewer #2: Partly

2. Has the statistical analysis been performed appropriately and rigorously? 

Reviewer #1: Yes

Reviewer #2: Yes

3. Have the authors made all data underlying the findings in their manuscript fully available?

Reviewer #1: Yes

Reviewer #2: Yes

4. Is the manuscript presented in an intelligible fashion and written in standard English?

Reviewer #1: Yes

Reviewer #2: Yes

5. Review Comments to the Author

Reviewer #1: General comments:

I think the manuscript is well written and the topic of great interest across different fields.

Yet, there are a few issues that in my view require further explanations.

Specific comments:

1. Abstract

I suggest that study aims be more precisely written because they are “somehow lost” in the “background objectives” section.

I did not clearly understand with there are cross-sectional samples in a longitudinal study. But I will come to this issue later on.

Results are somehow misleading, and not clearly enough to show their clinical relevance, although I know it is not always easy to write this in the abstract. Yet, I think authors can do a better job.

Finally, the “discussion conclusion” section ends up with what may be considered “common knowledge”. Can you please re-phrase it differently?

2. Introduction

It is well written, and apparently clear. I think it would benefit from a more substantive approach of the subject.

What do authors mean by “longitudinal stability”. This expression is often used in the draft, but no definition is given nor which statistic best captures this expression.

In paragraph two, in the last sentence, no clear explanation is given why further research is needed. Please explain.

The following paragraph also “suffers”, in my view, on too much descriptives instead of going a little deeper and suggest/explain/describe putative mechanism behind your correlates – socio-demographics and ethnicity. The same can be said in the last paragraph about cohort effects.

Finally, what are exactly the study aims? Are there any reasons why you did not formally posit hypotheses to be tested? If you provide these, please include also some substance linked to them.

Again, what is meant by longitudinal stability, and “would inform us on future trends“ (line 88). Are you thinking about prediction? If so, be more precise about its meaning and “clinical/intervention” implications.

3. Methods

Please provide more information about the LASA study, especially its aims and goals.

What about missing data and its putative implications in data analyses done?

Why do you include cross-sectional data in a longitudinal data set? Please explain, especially why include subjects with only one observation when you are interested in longitudinal stability!

At baseline having subjects with a 10 year lag (55-65) is problematic. Can you please explain why you use a decade lag.

In the “birth cohort, educational level and ethnic groups” entry birth cohorts were born before or during the 2nd World War with devastation consequences in the Netherlands. Yet, you did not mention this and its putative consequences, or is this issue of no relevance?

Statistical analyses are OK, although: (1) why do you adjust your analysis for height? Please explain; (2) what is this “longitudinal stability”?

Further down you wrote (line 159) “the modification of the stability, …” – what do you mean by this?

4. Results

Much of your results are written down, but the potential reader of the paper has no direct access to your results. I suggest, whenever needed, to include supplementary Tables so that the reader may “judge” by her/himself about them.

On the entre of “Longitudinal stability …” please be more precise. How do we know about it? Have all subjects the same “stability” or are there substantial differences? Please comment on this.

In the Modification by birth cohort entre, you write “there was no clinically relevant,…”. Yet, you never mention how can one decide about what is or is not clinical relevant. Are there cut-points for a reader to judge by her/himself?

In the Modification by educational level entry, I wonder if providing more detail (maybe a supplementary table) would help the reader to gain more insight about the results.

The very same suggestion goes to the Modification by ethnic groups entry.

5. Discussion

My main concern is that you do not provide substantive reasons/mechanisms/explanations in clinical terms about your findings. I think that this adding would increase the quality of the paper.

Reviewer #2: The authors investigate the sex differences in physical aging, which is addressed by the four measures gait speed, chair stand speed, handgrip strength, and balance. Several studies investigated the sex differences in different physical aging but only a few analyzed several measures and longitudinal data. This paper addresses the interesting question of potentially diminishing sex differences in physical aging. Overall, even though I see potential for this work, some issues need to be addressed.

Major issues:

The authors hypothesize that sex differences in physical performance diminish with increasing age, but what’s the expected mechanism? Why should the sex difference diminish and why should men and women perform at the same level? Several medical studies show differences in the body composition which is also related to muscle power relevant for e.g handgrip strength (next to hand size). The authors need to elaborate their motivation and the potential mechanism a bit more precise. Further why should a higher value in physical performance automatically go together with better health? There are also shown differences in body composition across continents (e.g. Europe and Asia).

Many studies on physical performance also showed body weight to be a relevant factor next to and in addition to body height. This measure should be included.

The authors have chosen a random intercept model to analyze the panel data. They call it random intercept for age, which is a very misleading term. Why was this model applied what is the advantage of this model here? Further, the statistical models should be formalized or at least the results should be provided within proper tables (either within the manuscript or as supplementary material). Right now, the model specification in steps 1 and 2 are not clear.

Another important issue: the description of the results is sparse at the moment. The authors refer quite often to Figure 1 when describing their results but they mis to provide numbers to support their visual interpretation. (In addition, the quality of Figure 1 makes it impossible to support all their interpretation). The interaction effect of age and sex is very relevant for this study, therefore I am quite surprised that there is no statistical result provided to show or disprove that sex differences are diminishing at higher ages.

Minor issues:

Was age centered within the analyses? (e.g around 55?)

The authors refer to other studies on younger adults showing no sex differences. This does not represent the full literature e.g https://doi.org/10.1371/journal.pone.0163917 shows sex differences in handgrip strength at young adulthood.

About the sample population:

An overview of dropouts per cohort group by sex might be relevant for readers to fully understand the sample. Some information is provided within the discussion section, but this needs to be done much earlier.

The high education subpopulation among the migration cohort is very small, authors need to consider dropping this group or interpret it with caution.

Table 1 could/should also include some information about participants refusing to perform the test or not willing to perform the test. Can these two categories be distinguished within the sample? How are participants treated who were willing but failed to perform the test or used the arms for the chair stand test….?

A limitation that should be considered when comparing the results: The chair stand test was not a maximum performance test as were the other three tests.

Was body height self-reported or measured?

References 11 and 24 are equal.

The quality of Figure 1 is poor .

6. PLOS authors have the option to publish the peer review history of their article (what does this mean?). If published, this will include your full peer review and any attached files.

**Do you want your identity to be public for this peer review?** For information about this choice, including consent withdrawal, please see our Privacy Policy.

Reviewer #1: No

Reviewer #2: No

We would appreciate receiving your revised manuscript by February, 2020. To enhance the reproducibility of your results, we recommend that if applicable you deposit your laboratory protocols in protocols.io, where a protocol can be assigned its own identifier (DOI) such that it can be cited independently in the future. For instructions see: http://journals.plos.org/plosone/s/submission-guidelines#loc-laboratory-protocols

We look forward to receiving your revised manuscript.

Kind regards,

Stephen D. Ginsberg, Ph.D.

Section Editor

PLOS ONE

2. During your revisions, please note that a simple title correction is required: the semicolon should be replaced by a colon. Please ensure this is updated both in the manuscript file and the online submission information.

3. Please provide additional details regarding participant consent.

In the ethics statement in the Methods and online submission information, please ensure that you have specified what type of consent you obtained (for instance, written or verbal).

If you obtained verbal consent, please state why it was not possible to obtain written consent, how verbal consent was recorded and whether the ethics committee approved this consent procedure.

4. Please include additional information regarding the survey or questionnaires used in the study and ensure that you have provided sufficient details that others could replicate the analyses. For instance, if you developed a questionnaire as part of this study and it is not under a copyright more restrictive than CC-BY, please include a copy, in both the original language and English, as Supporting Information.

6. Please include captions for your Supporting Information files at the end of your manuscript, and update any in-text citations to match accordingly. Please see our Supporting Information guidelines for more information: http://journals.plos.org/plosone/s/supporting-information

---

## [Author Response · Author response to Decision Letter 0]

23 Oct 2019

PONE-D-19-18300 L.D. Sialino et al. Response to Reviewers

We thank the reviewers for their comments. We made the suggested revisions in the manuscript. All revisions in the manuscript are highlighted by track-changes (Revised Manuscript with Track Changes). Our responses to the comments are written below, with references to lines in the final revised manuscript (Manuscript). 

Reviewer #1

I think the manuscript is well written and the topic of great interest across different fields. Yet, there are a few issues that in my view require further explanations. Specific comments:

1. Abstract: I suggest that study aims be more precisely written because they are “somehow lost” in the “background objectives” section. Results are somehow misleading, and not clearly enough to show their clinical relevance, although I know it is not always easy to write this in the abstract. Yet, I think authors can do a better job. Finally, the “discussion conclusion” section ends up with what may be considered “common knowledge”. Can you please re-phrase it differently?

We rewrote the Background and Objectives and Discussion and Conclusion part of the abstract. We included the study aims (lines 26 to 28) and wrote a more detailed conclusion with extra information on implications for practice to emphasize the clinical relevance (lines 43 to 48). 

2. Introduction: It is well written, and apparently clear. I think it would benefit from a more substantive approach of the subject. What do authors mean by “longitudinal stability”. This expression is often used in the draft, but no definition is given nor which statistic best captures this expression. Again, what is meant by longitudinal stability, and “would inform us on future trends“ (line 88). Are you thinking about prediction? If so, be more precise about its meaning and “clinical/intervention” implications.

Longitudinal stability referred to the stability by age. We used this term to refer to the course of the sex difference by age, also the stability of the sex difference. To prevent confusion, we changed the term into “longitudinal course by age” (line 55). The statistics behind this term is the sex*age and sex*age*age interaction, which is explained in an extra sentence added to the method (lines 179 to 181). In general, trends in health or disease in birth cohorts or generations in the past are often predictive for future trends. The two birth cohorts will indicate a stable, decrease or increase of the sex difference in physical performance. This provides an indication of whether the female disadvantage is an increasing or decreasing phenomenon within our changing society. It may also inform us whether this needs intervention in the future. A more detailed explanation was added (lines 94 to 99).

3. Introduction: In paragraph two, in the last sentence, no clear explanation is given why further research is needed. Please explain.

We added more information on the follow-up time of the described study (line 62 to 65). Further research with a longer follow-up time is needed to confirm the longitudinal finding (so far only a follow-up of 10 years) that the sex difference in gait speed remains stable during aging. In addition, we added a sentence on the practical implications of investigating this longitudinal course by age to better emphasize the relevance (line 64 to 66).

4. Introduction: The following paragraph also “suffers”, in my view, on too much descriptives instead of going a little deeper and suggest/explain/describe putative mechanism behind your correlates – socio-demographics and ethnicity. The same can be said in the last paragraph about cohort effects.

We split the paragraph on socio-demographics into two separate paragraphs (education and ethnicity). We now discussed in greater with more detailed the suggested putative mechanism behind the association between education/ethnicity and physical performance and our hypothesis why this might influence men and women differently. Please see the revised paragraph three and four of the introduction. The last paragraph about cohort effects was also extended with explanatory sentences (lines 90 to 94).

5. Introduction: Finally, what are exactly the study aims? Are there any reasons why you did not formally posit hypotheses to be tested? If you provide these, please include also some substance linked to them.

We added the study aims (lines 107 to 108). Furthermore, we posit sub-hypothesis based on previous literature within each paragraph of the introduction as was suggested by the reviewer. See the revised introduction.

6. Methods: Please provide more information about the LASA study, especially its aims and goals.

We revised the paragraph “Study population” of the methods accordingly and included a reference to the cohort paper of LASA (line 117).

7. Methods: What about missing data and its putative implications in data analyses done?

In our models we include sex, age and the outcome (physical performance). There are no missing values for sex and very limited for age (since it can be calculated by the date of the interview and birth date). So, there is only a missing when the whole interview was not carried out. Furthermore, LASA has a very low drop-out, the latter was a maximum of only 3% per wave. Both mixed models and Generalized Estimated Equations (GEE) use all longitudinal data of the outcome, not only the full case models, so this gives no problem for missing values for physical performance. Together with the fact that the missing values are limited (see Supplementary Table 1) we expect no implications due to missing values in our analysis. We agree that this should be made more clear in the method section and therefore added a paragraph on missing with reference to the paragraph statistical analyses (lines 169 to 175). Furthermore, to provide more insight into the missing values we included a supplementary table 1 and 2. These show the number of participants and its percentage of valid measurements versus the total, for each cohort, for each follow-up measurement, for each outcome measurement and for both men and women. 

8. Methods: Why do you include cross-sectional data in a longitudinal data set? Please explain, especially why include subjects with only one observation when you are interested in longitudinal stability!

The migration cohort including participants of Turkish and Moroccan ethnicity started in 2012/2013, with one measurement so far. These is unique data to study the influence of ethnicity on the sex difference in physical performance, since this is the first time such extensive data was collected in these ethnic groups in the Netherlands. Although it is cross-sectional data, it was the only way to investigate the influence of ethnicity, so we decided to use this cross-sectional data for a sub-analysis. We agree that this is not useful for studying a longitudinal course. Therefore, we only studied whether the sex difference was significantly different between different ethnic groups (sex*ethnicity) and not study the longitudinal course by age (sex*age*ethnicity). To explain this more clearly, we revised the sentences explaining the cross-sectional analyses throughout the analyses steps.

9. Methods: At baseline having subjects with a 10 year lag (55-65) is problematic. Can you please explain why you use a decade lag.

We agree that having subjects aged 55-65 years at baseline creates a cross-sectional part in a longitudinal analysis. We however do not see this as problematic per se. Our analyses are adjusted for age, the distribution of age in our participants over these 10 years is homogeneous and the follow-up time is longer than this baseline lag (24 years for birth cohort 1927-37 and 12 years for 1937-47). LASA contains data with a 10 year lag and we decided to include all these participants to ensure a high power (n ~ 1000) and comparability between LASA cohorts (all have a baseline of 55-65 years). 

10. Methods: In the “birth cohort, educational level and ethnic groups” entry birth cohorts were born before or during the 2nd World War with devastation consequences in the Netherlands. Yet, you did not mention this and its putative consequences, or is this issue of no relevance?

We hypothesized that the war might be of influence on the access to education. Indeed, we see that the birth cohort born during World War 2 has a lower percentage of high educated persons for both men and women. However, we do not expect that it affected men and women differently. Also, we showed in this study that education does not influence the sex difference in physical performance. It was investigated using the LASA data if the famine that took place during the war had an influence on the height and chance of developing diabetes and cardiovascular disease as older adults (F.R.M. Portrait et al. 2011 & 2017). It was demonstrated that these associations are significant for women, but not for men, which was suggested to be related to the fact that female health is more influenced by childhood events compared to men. We however, in our study, found no difference in effect on physical performance measures. We added a sentence relating to this point to the discussion (lines 303 to 306).

11. Methods: Statistical analyses are OK, although: (1) why do you adjust your analysis for height?

It has been demonstrated that sex differences in physical performance are partly explained by the height differences between men and women on a population level (see reference 65). Height affects the performance on the physical performance tests but is not related to the causal pathways between physical performance and various health outcomes. Therefore, we adjust for height to estimate the sex difference in physical performance without the bias of height in the performance measurement. Although body weight was also shown to partly explain the sex difference in physical performance, this is usually a consequence of life style patterns (diet and physical activity), which is in the causal pathway between physical performance and health outcomes. Therefore, we only adjusted our analyses for height. We added a sentence to the method section “statistical analyses” to explain this more elaborately (lines 192 to 197).

12. Methods: Please explain; (2) what is this “longitudinal stability”?

Longitudinal stability referred to the stability by age. We used this term to refer to the course of the sex difference by age, the stability of the sex difference. To prevent confusion, we changed the term into “longitudinal course by age” (line 55).

13. Methods: Further down you wrote (line 159) “the modification of the stability, …” – what do you mean by this?

We meant whether the course of the sex difference by age (increase, decrease or stable sex difference by age) was modified by birth cohort or educational level. In other words, whether the course of the sex difference was different for different birth cohorts or educational levels. We revised the sentence and added more explanation (lines 191 to 192). 

14. Results: Much of your results are written down, but the potential reader of the paper has no direct access to your results. I suggest, whenever needed, to include supplementary Tables so that the reader may “judge” by her/himself about them.

In agreement with this comment, we provided an extra Table 3 which shows the multivariate model of the longitudinal course of the height-adjusted sex difference per physical performance measure by age, for both longitudinal birth cohorts 1927-1937 and 1937-1947. In addition, an extra Supplementary Table 3 was provided including all the effect-modification results. 

15. Results: On the entre of “Longitudinal stability …” please be more precise. How do we know about it? Have all subjects the same “stability” or are there substantial differences? Please comment on this.

We tested the longitudinal course of the sex difference in physical performance by age by testing the interaction terms sex*age and sex*age*age (methods). The current study is a population study so we can only conclude that on a population level, the sex difference in average physical performance remains stable (or decreases for handgrip strength). It was not our objective to study and report on the individual differences in longitudinal stability. 

16. Results: In the Modification by birth cohort entre, you write “there was no clinically relevant,…”. Yet, you never mention how can one decide about what is or is not clinical relevant. Are there cut-points for a reader to judge by her/himself?

We discuss the cut-off points for clinical relevance for gait speed and handgrip strength in the discussion section. We agree that this should not be mentioned in the result section but only in the discussion section. We revised it accordingly. 

17. Results: In the Modification by educational level entry, I wonder if providing more detail (maybe a supplementary table) would help the reader to gain more insight about the results. The very same suggestion goes to the Modification by ethnic groups entry.

We provided Supplementary Table 3 which shows all tested effect modification by education, birth cohort and ethnicity with regression coefficient, 95% confidence interval and p-value. 

18. Discussion: My main concern is that you do not provide substantive reasons/mechanisms/explanations in clinical terms about your findings. I think that this adding would increase the quality of the paper.

The objective of this study was not to study the explanatory factors of the sex difference in physical performance, but provide a detailed understanding of the sex difference. In our discussion we do mention two possible explanatory theories (physiological or healthy aging). Since we do not test explanatory factors, we only briefly introduce the main factors from literature so far. We added an example of a sex difference in chronic disease osteoarthritis, that has been suggested to explain the observed sex difference in physical performance (lines 326 to 328). In future studies of our project “Sex and Aging” we will investigate explanatory factors. 

Reviewer #2

The authors investigate the sex differences in physical aging, which is addressed by the four measures gait speed, chair stand speed, handgrip strength, and balance. Several studies investigated the sex differences in different physical aging but only a few analyzed several measures and longitudinal data. This paper addresses the interesting question of potentially diminishing sex differences in physical aging. Overall, even though I see potential for this work, some issues need to be addressed.

Major issues: 

1. The authors hypothesize that sex differences in physical performance diminish with increasing age, but what’s the expected mechanism? Why should the sex difference diminish and why should men and women perform at the same level? Several medical studies show differences in the body composition which is also related to muscle power relevant for e.g handgrip strength (next to hand size). There are also shown differences in body composition across continents (e.g. Europe and Asia). The authors need to elaborate their motivation and the potential mechanism a bit more precise.

We understand the questions mentioned by the reviewer. We hypothesize that if the sex difference is stable by age, it suggests a physiological difference. We think the body composition is a potential explanatory factor here. This indeed does not mean that men and women have the same health at the same performance level, as was questioned by the reviewer. If the same health by men and women is measured at different physical performances, sex-specific cut-offs for the relationship between physical performance and health are needed. However, this needs to be assessed in future research. To discuss these points, we added two sentences on this topic (lines 312 to 314). Next to an physiological difference, the sex difference might be due to difference in (un)healthy aging life style factors. These factors might change of time, resulting in a change of the sex difference by age. However, all discussed points are not addressed in our research question and are therefore only discussed as possible theories in the discussion (lines 315 to 321)

2. Further why should a higher value in physical performance automatically go together with better health? 

There has been extensive research performed investigating the relationship between physical performance and various health outcomes. A strong association for handgrip strength, gait speed and lower body strength (party measured by chair stand) with probability of disability was observed. Also, gait speed and mortality were found to be associated. Although this does not mean that a person with higher physical performance always has a better health, it means that on average good physical performance is an indicator for health. To emphasize this, we only used the word “associated” and not “affects” or “means” in the introduction (lines 57 to 58).

3. Many studies on physical performance also showed body weight to be a relevant factor next to and in addition to body height. This measure should be included.

Height affects the performance on the physical performance tests but is not incorporated in the (hypothesized) causal pathway between physical performance and various health outcomes. Therefore, we adjust for height to estimate the sex difference in physical performance without the bias of height in the measurement of performance. Although body weight was also shown to partly explain the sex difference in physical performance, this is usually affected by life style (diet and physical activity), which may be part of the causal pathway between physical performance and health outcomes. Therefore, to show the unadjusted sex difference in physical performance, we only adjusted our analyses for height. We added a sentence to the method section “statistical analysis” to explain our choice more elaborately (lines 193 to 196).

4. The authors have chosen a random intercept model to analyze the panel data. They call it random intercept for age, which is a very misleading term. Why was this model applied what is the advantage of this model here? Further, the statistical models should be formalized or at least the results should be provided within proper tables (either within the manuscript or as supplementary material). Right now, the model specification in steps 1 and 2 are not clear.

We meant a random intercept for the individual, we revised the sentence accordingly. The advantage of this model is that it takes the correlation between the measurements within one person into account. To provide more insight in our statistical models we added Table 3, which shows our multivariable model and its build-up step by step as described in the method section.

5. Another important issue: the description of the results is sparse at the moment. The authors refer quite often to Figure 1 when describing their results but they mis to provide numbers to support their visual interpretation. (In addition, the quality of Figure 1 makes it impossible to support all their interpretation). The interaction effect of age and sex is very relevant for this study, therefore I am quite surprised that there is no statistical result provided to show or disprove that sex differences are diminishing at higher ages.

We added Table 2 to provide the statistical method used to produce figure 1 and show the interaction effects between age and sex. In addition, we also added Supplementary Table 3 to show the results of the effect modification analyses. 

Minor issues

1. Was age centered within the analyses? (e.g around 55?)

Age was not centered in our analysis. We decided to not center within the analysis because we only take age and sex as covariates in our analysis (where the value zero is not necessarily arbitrary). Also, we are mostly interested in the interaction terms, where age centered analysis do not have an beneficial effect.

2. The authors refer to other studies on younger adults showing no sex differences. This does not represent the full literature e.g https://doi.org/10.1371/journal.pone.0163917 shows sex differences in handgrip strength at young adulthood.

We adjusted the sentence in the discussion and added the reference advised by the reviewer (line 320). 

3. An overview of dropouts per cohort group by sex might be relevant for readers to fully understand the sample. Some information is provided within the discussion section, but this needs to be done much earlier.

We included two supplementary tables (1 and 2), which show the number of participants and its percentage of valid measurements versus the total, for each cohort, for each follow-up measurement, for each outcome measurement and for both men and women. 

4. The high education subpopulation among the migration cohort is very small, authors need to consider dropping this group or interpret it with caution.

We only analyze the migration cohort as a whole group together to estimate the sex difference and compare the sex difference with the Dutch cohort 1947-1957. Since we do not group by education level, we did not encounter a low power problem. 

5. Table 1 could/should also include some information about participants refusing to perform the test or not willing to perform the test. Can these two categories be distinguished within the sample? How are participants treated who were willing but failed to perform the test or used the arms for the chair stand test….?

We included supplementary tables 1 and 2 for an overview of the participants who refused, were not willing to, did not perform any interview or unknown missing values. Table 1 “unable” shows the participants who were unable to perform the test correctly or completely or refused due to being physically unable to perform the test (for example; participants in a wheelchair). To make this point more clear, we added an extra footnote to Table 1. 

6. A limitation that should be considered when comparing the results: The chair stand test was not a maximum performance test as were the other three tests.

The gait speed and handgrip strength test are maximum performance tests, but the balance and chair stand test are not. We do not see this as a limitation when comparing results. Since we are interested in the functioning these tests represent and not the maximum performance, both test are applicable. In addition, our goal is to detect a differentiation between participants and within in a participant by age, which is achieved by both our maximum and non-maximum performance tests. We however agree that the point made by the reviewer is something to take into account when interpreting the results. We therefore explained all measures in detail in the method section.

7. Was body height self-reported or measured?

Body height was measured to the nearest 0.001 m using a stadiometer by a trained interviewer. We added this information to the method section (lines 196 to 197).

8. References 11 and 24 are equal.

We removed the duplicate reference. 

9. The quality of Figure 1 is poor .

We used another program to make the matrix graph to ensure high quality after submission. Please see revised Figure 1.

Concluding author remarks

- All analyses were rerun in a newer version of SPSS (26) to create the additional tables. Therefore, the figures and values in tables and text slightly changed after the revision. 

- During the rerun of all analysis, we discovered a minor error in the gait speed analysis of LASA birth cohort 1927-1937. As a result, the effect sizes for the sex difference in gait speed were revised (see Table 1). This did not influence our final conclusions. 

- The graph for depicting the balance outcome measure was changed using the full model estimated regression coefficients instead of the raw data. This ensured a line that better fits to the model as was previously used.

---

## [Decision Letter · Decision Letter 1]

26 Nov 2019

Sex differences in physical performance by age, birth cohort, educational level and ethnic groups: The Longitudinal Aging Study Amsterdam

PONE-D-19-18300R1

Dear Dr. Sialino,

We are pleased to inform you that your manuscript has been judged scientifically suitable for publication and will be formally accepted for publication once it complies with all outstanding technical requirements.

With kind regards,

Stephen D. Ginsberg, Ph.D.

Section Editor

PLOS ONE

**Comments to the Author**

1. If the authors have adequately addressed your comments raised in a previous round of review and you feel that this manuscript is now acceptable for publication, you may indicate that here to bypass the “Comments to the Author” section, enter your conflict of interest statement in the “Confidential to Editor” section, and submit your "Accept" recommendation.

Reviewer #1: All comments have been addressed

Reviewer #2: All comments have been addressed

2. Is the manuscript technically sound, and do the data support the conclusions?

Reviewer #1: Yes

Reviewer #2: Yes

3. Has the statistical analysis been performed appropriately and rigorously? 

Reviewer #1: Yes

Reviewer #2: Yes

4. Have the authors made all data underlying the findings in their manuscript fully available?

Reviewer #1: Yes

Reviewer #2: No

5. Is the manuscript presented in an intelligible fashion and written in standard English?

Reviewer #1: Yes

Reviewer #2: Yes

6. Review Comments to the Author

Reviewer #1: I am satisfied with author´s answers to my comments/questions/suggestions. Further, the new draft is richer and more reliable.

I have no further questions.

Reviewer #2: Dear authors,

My raised issues were fully addressed. The manuscript approved a lot and reads well, in particular the methods and results section are now much clearer.

7. PLOS authors have the option to publish the peer review history of their article (what does this mean?). If published, this will include your full peer review and any attached files.

Reviewer #1: No

Reviewer #2: No

---

## [Editor Report · Acceptance letter]

10 Dec 2019

PONE-D-19-18300R1 

Sex differences in physical performance by age, educational level, ethnic groups and birth cohort: The Longitudinal Aging Study Amsterdam 

Dear Dr. Sialino:

I am pleased to inform you that your manuscript has been deemed suitable for publication in PLOS ONE. Congratulations! Your manuscript is now with our production department. 

With kind regards,

on behalf of

Dr. Stephen D Ginsberg 

Section Editor

PLOS ONE